# The Antimicrobial Resistance (AMR) Rates of Uropathogens in a Rural Western African Area—A Retrospective Single-Center Study from Kpando, Ghana

**DOI:** 10.3390/antibiotics11121808

**Published:** 2022-12-13

**Authors:** Susanne Deininger, Therese Gründler, Sebastian Hubertus Markus Deininger, Karina Lütcke, Harry Lütcke, James Agbesi, Williams Ladzaka, Eric Gyamfi, Florian Wichlas, Valeska Hofmann, Eva Erne, Peter Törzsök, Lukas Lusuardi, Jan Marco Kern, Christian Deininger

**Affiliations:** 1Department of Urology and Andrology, Salzburg University Hospital, Paracelsus Medical University, 5020 Salzburg, Austria; 2Doctors for Africa e. V., 77654 Offenburg, Germany; 3No Limit Surgery (NLS), 5020 Salzburg, Austria; 4Margret Marquart Catholic Hospital, Kpando, Ghana; 5University Clinic of Urology, Eberhard Karls University, 72076 Tübingen, Germany; 6University Institute of Clinical Microbiology and Hygiene, Paracelsus Medical University, 5020 Salzburg, Austria; 7Institute of Tendon and Bone Regeneration, Spinal Cord Injury & Tissue Regeneration Center Salzburg, Paracelsus Medical University, 5020 Salzburg, Austria

**Keywords:** low-income country, uropathogen infection, bacterial resistance rates

## Abstract

Little is known about the antimicrobial resistance (AMR) status of uropathogens in Western Africa. We performed a retrospective evaluation of urine cultures collected from the rural Margret Marquart Catholic Hospital, Kpando, Ghana during the time period from October 2019–December 2021. Urine samples from 348 patients (median age 40 years, 52.6% male) were examined. Of these, 125 (35.9%) showed either fungal or bacterial growth, including *Escherichia coli* in 48 (38.4%), *Candida species* (spp.) in 29 (23.2%), *Klebsiella* spp. in 27 (21.6%), *Proteus* spp. in 12 (9.6%), *Citrobacter* spp. in 10 (8.0%), *Salmonella* spp. in 4 (3.2%), *Staphylococcus* spp. in 3 (2.4%), and *Pseudomonas* spp. in 2 (1.6%) cases. Two bacterial spp. were detected in 7 samples (5.6%). Antibiotic susceptibility testing showed resistance to a mean 8.6 out of 11 tested antibiotics per patient. Significant predictors (*p* < 0.05) of bacterial growth were age (OR 1.03), female sex (OR 3.84), and the number of pus cells (OR 1.05) and epithelial cells (OR 1.07) in urine microscopy. We observed an alarmingly high AMR rate among the uropathogens detected, even to reserve antibiotics. A similar resistance profile can be expected in West African patients living in high-income countries. These observations warrant the implementation of restrictive antibiotic protocols, together with the expansion of urine culture testing capacities, improvement of documentation and reporting of AMR rates, and continued research and development of new antibiotic therapies in order to stem the progression of AMR in this West African region.

## 1. Introduction

The World Health Organization (WHO) defines antimicrobial resistance (AMR) as a “change of bacteria, viruses, fungi and parasites so that they no longer respond to medicines, making infections harder to treat” [1]. The global increase in AMR is a serious problem in the treatment of infectious diseases. Nevertheless, the phenomenon of AMR was recognized before the discovery of the first antibiotic, penicillin, in 1928 by Sir Alexander Fleming. Salvarsan (ingredient: Arsphenamine) was a synthetic drug used for the treatment of syphilis until the widespread use of penicillin for this indication [2], and AMR of the causative pathogen *Treponema pallidum* was described soon after its introduction in clinical therapy [2,3]. Besides natural innate resistance mechanisms such as penicillinases (discovered in 1940) [4], acquired resistance mechanisms were also discovered in bacteria as early as the 1940s.

AMR can develop via several mechanisms. Mutations may arise in the antibiotic’s target, as with fluorchinolone resistance, which is typically due to mutations in DNA-topoisomerase [5]. Alternatively, the drug can be enzymatically inactivated, as is the case with the numerous forms of betalactamases, aminoglycoside inactivation via phosphorylation, acetylation or adenylation. Two further mechanisms include bypassing of the target (e.g., shown in vancomycin AMR) and inhibiting access of the drug to its specific targets [6], which can be seen in tetracycline (TET) AMR (compare [7,8]).

Currently, the WHO refers to AMR as “one of the biggest threats to global health, food security, and development” [1]. Since 2015, the Global Antimicrobial Resistance and Use Surveillance System (GLASS) [9] has represented the WHO’s largest initiative for documenting global AMR and antibiotic use. With regards to uropathogenic germs, GLASS collects data on the AMR rates of *Escherichia (E.) coli* and *Klebsiella (K.) pneumonia*, with its 2021 report indicating a median AMR against cotrimoxazole of 54.4% (IQR 36.5–69.4) for *E. coli* and 43.1% (IQR 31.8–57.5) for *K. pneumonia* in 12 countries reporting worldwide. As of February 2021, 30 of 47 African countries completed their enrolment in the GLASS programme for AMR or antibiotic consumption (=AMC) surveillance. Nevertheless, data on AMR from some African regions remains scarce. In a 2017 systemic review by Tadesse et al. [10], only insufficient AMR data were available from 42.6% of 54 African countries, with only one study included from Ghana.

The German non-profit organisation Doctors for Africa e.V. [11] carries out humanitarian healthcare missions in urology several times a year at six different Ghanaian clinics that do not provide regular urological care. Since 2019, the organisation has been supporting the establishment of a microbiology department in the local laboratory of the Margret Marquart Catholic Hospital (MMCH) in Kpando, both financially and with specialised knowledge. Kpando is a rural town of 16.000 inhabitants, located in the Volta Lake region.

The purpose of this study was to systematically record the data on uropathogens from the local microbiology department, evaluate the bacterial load and AMR status, and to make evidence-based recommendations on calculated antibiotic therapies, e.g., in the context of perioperative antibiotic administration. The data are intended to supplement the currently inadequate data on the distribution and resistance status of uropathogens from the Western African region. When treating patients from these regions in high-income counties (HIC), a similar bacterial resistance profile can be expected and the information here used to guide treatment decisions.

## 2. Methods

Consent of the Salzburg ethics committee was obtained (registration number 1020/2022). Subsequently, a retrospective evaluation of the urine cultures performed by the local laboratory at MMCH over a time period of 27 months (10/2019–12/2021) was performed by the authors. The distance from the rural hospital in Kpando to the Ghanaian capital Accra is 192 km, but due to poor road conditions, the trip takes 5–6 h. The nearest major hospital is Ho Teaching Hospital, a 1.5-h drive away. This means that patients usually have to be cared for on site. The MMCH houses 200 inpatient beds, with the following specialties permanently represented: Internal Medicine, Gynecology, Pediatrics, Non-Surgical Ophthalmology, and Anesthesiology (Nursing). Elective surgical care is provided twice a week. Presently, the microbiology lab can analyze urine cultures, but also blood cultures and cultures drawn from vaginal swabs and wound swabs. Due to the absence of sophisticated electronic documentation at the MMCH, only an evaluation of data provided in a database was possible. Clinical data, such as complaints or the clinical course of the patients, could not be evaluated. Inclusion criteria: All patients who had been treated on an inpatient or outpatient basis during the above-mentioned period and had a urine culture prepared. Exclusion criteria: Missing data.

### 2.1. Local Procedure for Urine Microscopy and Establishment of Urine Cultures

Midstream-clean-catch urine was collected from patients directly in the local laboratory and processed immediately. For this reason, no stabilizer was added. The bacteria, erythrocytes, leukocytes, and yeast cells were counted by examining 100 high power fields (HPF) of the wet preparation of urine microscopically. Then the number of cells was averaged. Standard urine culture protocol used 1 µL of urine plated onto Cystine Lactose Electrolyte Deficient (C.L.E.D.) agar supplemented with Andrade indicator (manufacturer TECHNO PHARMGHEM, New Delhi, India) and incubated aerobically at 35–37 °C for 18–24 h (compare [12]).

### 2.2. Bacterial Identification and Count

Native urine cultures were primarily used for bacterial identification. Primarily, pathogen identification was performed by colony morphology and examination of the phenotype under the microscope [13]. The addition of the Andrade indicator within the C.L.E.D. agar allowed visual differentiation of bacterial species. The following color scheme was used for identification [14]:slight yellowish or greenish: *Enterococcus faecalis*yellow, opaque centre slightly deeper yellow: *E. coli*yellow to whitish blue: *K. pneumonia*Blue: *Proteus vulgaris*Bluish: *Salmonella typhi*Deep yellow: *Staphylococcus aureus.*

Gram staining supported species differentiation in case of unclear color morphology. [13,15]. If *Pseudomonas* spp. were suspected, an oxidase test (Bactident™ Oxidase test strips, manufacturer Merck) was used for confirmation. Differentiation between *Staphylococcus aureus* and coagulase-negative staphylococci was achieved using the coagulase test (self-produced from human plasma and peptone water from manufacturer TECHNO PHARMGEM, New Delhi, India, in a ratio of 1:4) [16]. Of note, documentation of bacterial count and differentiation between significant and insignificant growth was introduced into standard practice in December 2021 and was not noted until then.

The following bacteria were detected and have been classified according to their bacterial genus for simplicity. In some cases, no specific classification of a uropathogen into the exact species was made during clinical documentation; therefore, only the designated genus is provided (with the comment “not further classified” (NFC)):*Candida* spp.*Citrobacter koseri* and *Citrobacter* NFC: referred to as *Citrobacter* spp.*Corynebacterium**Escherichia* (E.) *coli**Klebsiella* (K.) *pneumonia*, *Klebsiella oxytoca* and *Klebsiella* NFC: referred to as *Klebsiella* spp.*Proteus vulgaris* and *Proteus mirabilis*: referred to as *Proteus* spp.*Pseudomonas aeruginosa* and *Pseudomonas* NFC: referred to as *Pseudomonas* spp.*Salmonella parathyphii* and *Salmonella* NFC: referred to as *Salmonella* spp.*Staphylococcus aureus* and *Staphylococcus epidermidis*: referred to as *Staphylococcus* spp.

### 2.3. Antibacteril Susceptibility

Susceptibility test of the bacterial isolates was performed on a Müller-Hinton-Agar (manufacturer HiMedia Laboratories Pvt. Ltd., Mumbai, India), with bacterial inoculum at 0,5 McFarland standard. The Kirby–Bauer disk diffusion test was used to determine the antibiotic susceptibilities. Cultural characteristics were observed after incubation at 30–35 °C for 18–24 h [17]. Zone diameters were interpreted referring to the CLSI 2015 breakpoints (CLSI, 2015 [18]).

Various antibiotic test rings were used throughout the study period, depending on market availability. Table 1 lists the antibiotics (AB) that were used in these analyses and were therefore included in the systematic evaluation.

Other ABs (trimethoprim plus sulfamethoxazole [SXT] and fosfomycin) were not tested as standard and were therefore not included in the evaluation. The last antibiotic ring used was the “BDR001-Urine isolates (12 discs ring)” (manufacturer Biomark^®^ Laboratories). This contained the antibiotics in the following concentrations: AMC 30 μg, CIP 5 μg, CRO 30 μg, GEN 10 μg, PIP 20 μg, NFN 300 μg, NAL 30 μg, CAZ 20 μg, NOR 20 μg, TET 30 μg, AMK 30 μg, and LEV 5 μg. Intermediate sensitivity to various antibiotics was found 36 times. These were marked as resistant to simplify data analysis.

### 2.4. Statistical Analysis

Statistical analysis was performed using the free software R. Specifically, we used a logit model to examine significant predictors of bacterial growth. Associations between socio-demographic characteristics, the findings of urine microscopy, and bacterial growth in the microbiological examination were assessed using the chi-square test. A *p*-value < 0.05 was considered statistically significant. Microsoft Excel was used to generate the graphs.

## 3. Results

### 3.1. Descriptive Statistics

During the time period from 10/2019–12/2021, urine cultures were performed on samples collected from N = 348 patients, of which 52.6% (n = 180) were male and 47.4% (n = 162) were female. Data on sex was not available for n = 6 patients. Out of n = 348 urine cultures, n = 131 (37.6%) showed microbial growth, with n = 110 (31.6%) showing bacterial growth, and n = 29 (8.3%) exhibiting fungal growth. Seven urine cultures (2.0%) showed the growth of two different uropathogens. In these cases, each uropathogen was counted as a separate event in the statistical analysis.

Basic characteristics of patients and results of urine microscopy and microbiology testing of patients with positive cultures can be found in Table 2.

### 3.2. Urine Culture Results

In n = 110 positive urine cultures, the following bacterial species were identified: *E. coli* in 42.7% (n = 47), *Citrobacter species* (spp.) in 24.5% (n = 27), *Salmonella* spp. in 10.9% (n = 12), *Pseudomonas* spp. in 9.1% (n = 10), *Proteus* spp. in 4.5% (n = 5), *K.* spp. in 3.6% (n = 4), *Staphylococcus* spp. in 2.7% (n = 3), and *Corynebacterium* spp. in 1.8% (n = 2) of positive samples.

The distribution of bacterial species identified in all n = 110 positive cultures is depicted in Figure 1.

### 3.3. AMR of Bacterial spp.

The AB susceptibility patterns to the various antibiotics tested are shown in Table 3 for all uropathogens detected in our study.

### 3.4. Clinical Predictors of Bacterial Growth

The statistical correlation between urine microscopy and subsequent bacterial growth in urine culture was investigated and the results are shown in Table 4.

Independent, significant predictors of bacterial growth in urine culture were age (OR 1.03), female sex (OR 3.84), and presence of pus cells (OR 1.05) and epithelial cells (OR 1.07) in urine microscopy.

## 4. Discussion

To our knowledge, this study currently represents the largest Ghanaian uropathogen database available that additionally includes comprehensive antibiotic susceptibility information. Previously published data on uropathogens in Ghana are less extensive than ours due to a variety of reasons or combinations thereof. For example, the data (i) are partially more than 10 years old, (ii) are based on small cohorts or very special sub-cohorts, such as pregnant women or diabetics, or (iii) were collected in a multi-center setting. Additionally, some studies have only tested susceptibility to CIP, or investigated only single or a few uropathogens. To better highlight our data against the existing literature, we list in Table 5 the AMR rates for *E. coli*, the most commonly detected uropathogen, and include literature reporting *E. coli* strains from urine and blood cultures from Ghana, Europe, Austria (Austrian resistance (AURES) reports from 2016 and 2020) and the United States of America (USA).

*E. coli* remains the predominant pathogen in UTI. However, AMR data of this study differs a lot compared to Austrian data: The overall resistance rate of *E. coli* (invasive bacteria) towards aminopenicillins ranges between 50.5% (2016) and 46.1% (2020). Resistance of third generation cephalosporins (which indicates the ESBL-rate) remains low (around 10% between 2016 and 2020), whereas resistance towards fluorochinolones varies between 19.8% (2016) and 17.8% (2020). The overall resistance rate against aminoglycosides is 6.4%. There is no relevant resistance against carbapenems in Austria (24 isolates which were sent for confirmation to the Austrian reference laboratory) [24].

In 2019, Donkor et al. prospectively examined N = 31 positive urine cultures from patients seen at several polyclinics in the Ghanaian capital of Accra [21]. This cohort is comparable to ours in age (median age 37.2 yrs [21] vs. 40 yrs) although the fraction of female study participants was significantly higher at 79.8% [21] compared to 47.4% in our cohort. The collective data indicate that urinary tract infection (UTI) patients in Ghana are younger compared to those in HICs as reflected in the literature. Comparing UTI patients from EU and non-EU countries in 2001, Bouza et al. observed that patients from the EU had a median age of 67.95 years, while those from non-EU countries had a median 52.54 years of age (*p* < 0.05) [26]. In our cohort, age was a significant predictor of bacterial growth in urine cultures (OR 1.03). For each year, the odds increased by 3 per cent. In contrast, a study including 90 diabetic Ghanaian patients analysed by Forson et al. in 2021 [19] demonstrated that the risk of bacterial growth decreased with increasing age (OR 0.417), and in the Bouza study [26], pregnancy emerged as the only clinical risk factor among study participants (OR 2.42). Sex, age, diabetes, history of UTI, and frequency of sex showed no significant association with UTI [26]. Of note, in our patient population, it was not possible to extract the clinical history of the patients. Because of this, certain correlations, e.g., that of pregnancy with UTI, could not be investigated. It was also not possible to clarify the symptoms of the patients on the basis of our data. It is therefore unclear whether it was a symptomatic UTI or a case of contamination.

As far as the distribution of uropathogen species is concerned, our data supports previously published Ghanaian data. *E. coli* was the most frequent bacterial species in both our study and the Donkor study [21], representing 42.7% and 48.4% of positive cases, respectively. However, while *K.* spp. (16.1%) and *Staph. aureus* (12.9%) followed *E. coli* in the Donkor study, in our cohort, *Citrobacter* spp. (24.5%) and *Salmonella* spp. (10.9%) were the next most frequent uropathogens. Possibly this could be a handling or species detection bias, as the spectrum of *Citrobacter*, *S. aureus* or *Salmonella* is not sex-dependent. The presence of risk factors, such as diabetes, potentially impacts the distribution of detected uropathogen. Studying a cohort of 90 Ghanaian diabetic patients with UTI, Forson et al. found that *K.* spp. (55.6%) was the predominant uropathogen identified, with *E. coli* coming in second at 31.3% [19].

Our study highlights extremely high AMR rates for all uropathogens detected, which appears to be even more pronounced in this rural Ghanaian area than in the capital city of Accra. In 2015, Afriyie et al. examined n = 112 positive urine cultures from the Ghana Police Hospital, Accra, but only tested for AMR against CIP. Most frequently, *E. coli* (46.4%) was isolated, which displayed AMR rates against CIP of 38.5% [22]. Likewise, in the 2021 Forson study conducted at the Korle- Bu Hospital in Accra, only 21.4% of *E. coli* strains showed resistance to CIP [19]. Notably, in our cohort, the resistance rate of *E. coli* to CIP was 89.4%. Similarly, resistance rates of *E. coli* strains (n = 15) to fluoroquinolones was reported at 20 (CIP)–40% (NOR) in the Donkor study from Accra [21], whereas in our study, 51.1% (LEV)–91.5% (NOR) of *E. coli* strains (n = 47) were resistant. In a 2018 study by Forson et al., which included 82 *E. coli* isolates from pregnant women, multiple antibiotic resistances were observed, with AMR rates to AMP at 79.3%, to TET, 70.7%, to cotrimoxazole, 59.8%, and to CXM, 32.9% [20]. Our data demonstrates high resistance rates of up to 100% to multiple standard antibiotics such as aminopenicillin +/− β-lactamase inhibitors, cephalosporines, or fluorochinolones, which, of course, must be distinguished from primary resistance. Of more immediate concern are the *E. coli* AMR rates against reserve antibiotics such as GEN and TET, which we report here to be at 61.7% (vs. 26.7% in Donkor et al. [21] and 14.3% in Forson et al. [19]) and 87.2% (vs. 26.7% in Donkor et al. [21]), respectively. It must also be borne in mind that GEN is a purely intravenous antibiotic. Therefore, its use is limited to healthcare facilities. This is certainly one of the reasons why the rate of AMR is even lower than for oral antibiotics. Only difficult-to-obtain drugs such as AMK still appear to demonstrate acceptable rates of efficacy against these multi-resistant uropathogens.

In the capital city of Accra, AMR to last-resort antibiotics, such as AMK, appear to have increased in recent years. While only 6.7% of *E. coli* strains were reported to be resistant to AMK in the 2019 Donkor study [21], this number rose to 21.4% in the 2021 Forson study. It is important to note, however, that study cohorts may vary significantly, due to low sample numbers and lack of clinical comparability. Our study reports that only 10.6% of *E. coli* strains at MMCH were resistant to AMK, which may be due to the fact that access to AMK is upon request at the pharmacy in Kpando, and was not available at the time of the retrospective analysis Furthermore, the Ghanaian public health system only covers certain antibiotics (including CIP, GEN, AMC, certain cephalosporins) while others (including AMK, LEV, TET, NOR, NAL, PIP, NFN) must be paid for in part or in full by the patients themselves. In addition, AMK is hardly used in HICs today. True “last resort” antibiotics here are Cefiderocol, Meropenem-vaborbactam, and Colistin [27]. However, these are hardly available in low-income countries (LICs). As a result, substances such as AMK are experiencing a renaissance. The causes of AMR are manifold. From an evolutionary biology point of view, the mechanism is clear: biological pressure from antibiotics selects individual pathogenic strains that have developed an AMR mechanism through genetic mutation. From a clinical standpoint, the European Centre for Disease Prevention (ECDC) has identified the misuse of ABs as one of the leading causes of AMR [28]. According to the ECDC, misuse occurs mainly in the following three scenarios: (i) unnecessary prescription of ABs for viral infections, (ii) prescription of broad-spectrum ABs as a result of ignorance of the causative pathogen, and (iii) incorrect doses, frequencies, or treatment durations of AB administration.

To prevent AMR, the ECDC suggests the following: restrained and correct application of ABs, and adopting of hygienic precautions for the control of cross-transmission of AMR pathogens [28]. Indeed, taking into account a study by Olu-Taiwo et al. from 2021, which carried out a microbial examination of mobile phones and computer keyboards of Ghanaian healthcare university students, hygienic precautions will become increasingly important for reducing local transmission rates. Microbial contamination of 83.3% of mobile phones and 43.3% of computer keyboards was found, including 12.9% *K.* spp. and 6.7% *E. coli*. Overall, high rates of resistance to AMP (96.7%) and TET (75.8%) were observed, and *E. coli* showed AMR rates of 18.6% [29]. Acolatse et al. also showed widespread contamination with ESBL- and carbapenemase- producing gram-negative bacteria in surface swabs from a tertiary Ghanaian hospital in 2022 [30].

However, an even more important aspect of preventing the development of AMR is continuous monitoring for bacterial species and their corresponding susceptibilities, as has been carried out at the MMCH for more than 2 years now. With this, the supra-regional health center contributes decisively to targeted therapy, care research, and documentation of the local AMR situation. Unfortunately, this is not yet possible across the board in other rural regions.

In urology, AB therapy appears to be unavoidable, especially in invasive procedures, due to the frequent contamination of the urinary tract. The guideline “Urological Infections” of the European Association of Urology (EAU) recommends perioperative AB prophylaxis for the following procedures: ureteroscopy, percutaneous nephrolithotomy, transurethral resection of the prostate, transurethral resection of the bladder in high-risk patients, and transrectal prostate biopsy [31]. Even in regions with lower AMR rates than in our cohort, there is the option, e.g., in the case of transrectal prostate biopsies, to take a rectal swab prior to intervention in order to administer the appropriate AB prophylactically [31]. In a patient cohort such as in this study, with AMR rates of up to 100% against standard ABs, such guidelines can hardly be implemented. Potentially, this may mean dispensing with protective perioperative AB administration, or only applying ABs after a prior urine culture. Clear treatment algorithms and standards could help to avoid treatment of urine contamination only, and so reduce the misuse of antibiotics. e.g., after application of a dipstick, urine microscopy could provide information about the presence of erythrocytes, leukocytes and bacteria in the urine. Even in the case of a symptomatic infection, supportive therapy, e.g., infusions, must primarily be carried out until the urine culture is available, provided that the patient’s clinical condition is acceptable. At least until identification of the uropathogen, which is usually possible after 1–2 days [32], the data from the present study allows a rough orientation of the local/regional AMR situation that might be used to guide treatment. As a primary and simplest analysis, urine microscopy alone can give good indication of subsequent bacterial growth. In our study, the number of pus cells (OR 1.05) and epithelial cells (OR 1.07) determined by microscopy were independent predictors of UTI.

Rising AMR rates are a significant problem worldwide. A study published in The Lancet in 2022 calculates 1.27 million deaths worldwide from bacterial AMR in 2019. This number could increase to 10 million deaths/year by 2050 [33]. In addition, our arsenal against this “invisible enemy” appears to be shrinking. In 2019, the WHO identified only 32 ABs or AB combinations with a new therapeutic entity that were in clinical development for WHO priority pathogens, of which only 6 were considered innovative [28]. Moreover, access to certain ABs is limited in some regions. In this respect, there needs to be a central effort by healthcare professionals and researchers worldwide to prevent the further development of AMR and to identify new therapeutic targets. Local measures, such as restrictive antibiotic protocols, in combination with expansion of urine culture testing capacities, country- and region-wide improvement of documentation and reporting of AMR rates, and international measures, such as continued research and development of new antibiotic therapies, are all needed to effectively halt this process.

### Limitations of the Study

This is a retrospective data analysis from a remote hospital in rural Ghana, Africa. Due to missing data, symptoms and complaints of the patients as well as further characteristics, e.g., whether they were outpatients or inpatients, could not be included in the analysis. Also, due to a glaring lack of resources, only a selected number of antibiotics could be tested and evaluated. The test disks used were from different manufacturers, depending on local availability. Due to this also summary of the cephalosporins, as the individual substances could not be identified with certainty in the documentation (abbreviation partly as “Cef” or similar). Other antibiotics could not be evaluated (e.g., SXT) because they were not present on all of the rings. The test modalities such as agar plates, incubation temperature and incubation time could have differed. Due to the limited equipment no technical standard differentiation technique has been used (biochemical, PCR or mass spectronomy). No distinction between significant and insignificant bacterial growth has been made, no bacterial count was recorded. Thus no clear differentiation between UTI and contamination possible. The head of the local laboratory was instructed by a physician from the “Doctors for Africa” team to perform the microbiological examinations like reading test results/ agar plates and inhibition zone. There is no certified microbiologist on site, and the laboratory staff demonstrate varying levels of knowledge regarding the examinations performed (staff bias). Due to the location and limited capacities, the equipment and maintenance of technical devices is not comparable with those in HIC. Nevertheless, the data collected, and the statistical analysis provide a glimpse of the resistance situation as well as the needs of patients in this part of the world.

## Figures and Tables

**Figure 1 antibiotics-11-01808-f001:**
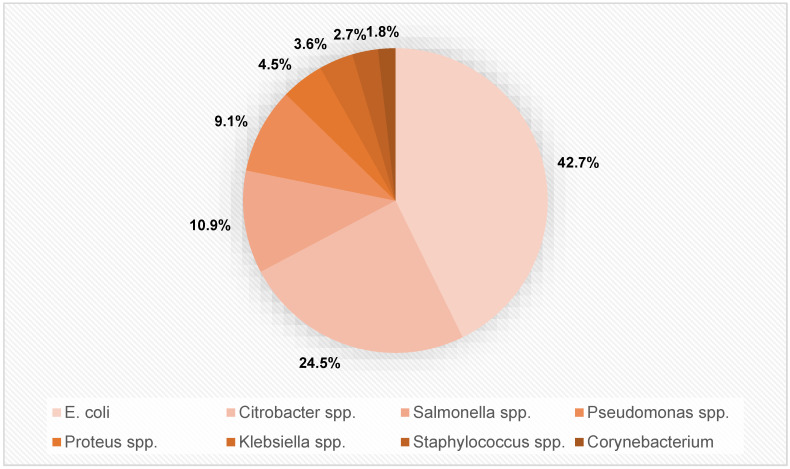
Distribution of bacterial species in n = 110 positive urine cultures in our study.

**Table 1 antibiotics-11-01808-t001:** List of antibiotics (AB) evaluated in antibiotic susceptibility tests.

Aminoglycosides	Gentamicin (GEN)
	Amikacin (AMK)
Aminopenicillins +/− ß-lactam inhibitor	Piperacillin (PIP)
	Amoxicillin + clavulanic acid (AMC)
Cephalosporins	Due to the different composition of the antibiotic rings, various cephalosporins are grouped together under this heading (including cefuroxime, cefazolin, and ceftriaxone). A separate evaluation was not possible because of the inconsistent documentation.
Fluorchinolones and Diazanaphthaline	Ciprofloxacin (CIP)
	Levofloxacin (LEV)
	Norfloxacin (NOR)
	Nalidixic acid (NAL)
Nitrofurantoin (NFN)	
Tetracyclines	Tetracycline (TET)

**Table 2 antibiotics-11-01808-t002:** Summary of patient characteristics and results of standard testing (n = 110).

		Mean	SD	Min.	Max.
Patients’ characteristics	Patients’ age in years	46.1	23.2	0	97
Urine microscopy: cells/high power field	Pus cells	13.0	27.0	0	250
	Epithelial cells	4.4	4.8	0	31
	Red blood cells	6.8	19.2	0	150
Urine microbiology	Total number of AMR per patient	8.6	2.1	2	11

MR = antimicrobial resistance, SD = standard deviation, Min. = Minimum, Max. = Maximum.

**Table 3 antibiotics-11-01808-t003:** Antibiotic susceptibility patterns of uropathogens in percent for all antibiotics tested.

Bacterial spp. (Number of Isolates)		Antibiotics										
		Aminoglykosides		Aminopenicillins +/−ß-Lactam AB		Cephalosporins	Fluorchinolones and Diazanaphthaline				NFN	TET
		AMK	GEN	AMC	PIP		CIP	LEV	NOR	NAL		
1	***Citrobacter* spp. (n = 27)**	0.0	66.7	/	100.0	88.9	74.1	55.6	81.5	77.8	85.2	77.8
2	***Corynebacterium* (n = 2)**	50.0	100.0	100.0	100.0	100.0	50.0	100.0	100.0	100.0	100.0	100.0
3	***Escherichia* *coli* (n = 47)**	10.6	61.7	100.0	100.0	85.1	89.4	51.1	91.5	97.9	55.3	87.2
4	***Klebsiella* spp. ° (n = 4)**	25.0	50.0	75.0	100.0	75.0	50.0	0.0	50.0	75.0	75.0	100.0
5	***Proteus* spp. * (n = 5)**	80.0	100.0	100.0	100.0	100.0	100.0	80.0	100.0	100.0	80.0	/
6	***Pseudomonas* (n = 10)**	0.0	80.0	/	100.0	/	100.0	60.0	100.0	100.0	70.0	80.0
7	***Salmonella* spp. (n = 12)**	0.0	66.7	100.0	91.7	83.3	66.7	41.7	83.3	75.0	66.7	83.3
8	***Staphylococcus* spp. (n = 3)**	0.0	66.7	100.0	100.0	100.0	66.7	33.3	66.7	100.0	66.7	66.7

AMK = Amikacin, AMC = Amoxicillin + clavulanic acid, CIP = Ciprofloxacin, GEN = Gentamicin, LEV = Levofloxacin, NAL = Nalidixic acid, NFN = Nitrofurantoin, NOR = Norfloxacin, PIP = Piperacillin, TET = Tetracycline; spp. = species. Colour coding: resistance rate 0%, 1–25%, 26–50%, 51–75%, 76–99%, 100%. Primary resistance: 
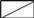
 ° comment on *Klebsiella* spp. susceptibility: susceptibility of AMC but PIP resistance is questionable. * comment on *Proteus* spp. susceptibility: swarming phenomenon on disc diffusion AMR testing might be interpreted as resistant but might have shown an “outer zone diameter”. This might explain the high R rate in *Proteus* spp.

**Table 4 antibiotics-11-01808-t004:** Statistical correlation between endogenous variables and microbial growth in urine cultures (HPF = high power field).

Estimation	(1)		(2)		(3)		(4)	
Endog. Variable	Bacteria Growth							
Predictors	OR	SE	OR	SE	OR	SE	OR	SE
**Intercept**	0.11 ***	0.48	0.07 ***	0.53	0.09 ***	0.50	0.11 ***	0.49
**Age in yrs**	1.03 ***	0.01	1.03 ***	0.01	1.03 ***	0.01	1.03 ***	0.01
**Sex (0: male, 1: female)**	2.88 *	0.61	3.84 **	0.66	2.31	0.63	2.52	0.63
**Age * Sex**	0.99	0.01	0.98	0.01	0.99	0.01	0.99	0.01
**Pus cells/HPF**			1.05 ***	0.01				
**Epithelial cells/HPF**					1.07 **	0.03		
**Red blood cells(HPF)**							1.01	0.01
**Observations**	333		320		322		322	
**R^2^ Tjur**	0.070		0.164		0.090		0.078	

* *p* < 0.1, ** *p* < 0.05, *** *p* < 0.01; OR: Odds ratio; SE: standard error; yrs: years; endog.: endogenous.

**Table 5 antibiotics-11-01808-t005:** Comparison of own data (Deininger et al., 2022) with existing literature on *E. coli* AMR rates in Ghana, Europe, the USA and the Austrian resistance (AURES) reports from 2016 and 2020.

Publication	Year	Origin of Specimen	Number of *E. coli* Isolates	Special Features of the Cohort	Antibiotics												
					Aminoglykosides		Aminopenicillins +/−ß-Lactam AB			Cephalosporins	Fluorochinolones and Diazanaphthaline				NFN	SXT	TET
					AMK	GEN	AMC	AMP	PIP		CIP	LEV	NOR	NAL			
Ghana																	
Deininger et al.	2022	Urine	47		10.6	61.7	100.0	NA	100.0	85.1	89.4	51.1	91.5	97.9	55.3	NA	87.2
Forson et al. [19]	2021		28	Diabetics	NA	14.3	21.4	85.7	NA	28.6 (CFX) 35.7 (CRO)	21.4	NA	NA	50	NA	42.8	NA
Forson et al. [20]	2018		82	Pregnant women	NA	41.5	NA	79.3	NA	32.9 (CMX)	NA	NA	NA	48.8	35.4	59.8	70.7
Donkor et al. [21]	2019		15		6.7	26.7	93.4	NA	93.4	26.7 (CAZ) 6.7 (CXM)	20.0	20.0	40.0	73.4	26.7	NA	53.4
Afriyie et al. [22]	2015		52		NA	NA	NA	NA	NA	NA	38.5	NA	NA	NA	NA	NA	NA
Europe																	
Critchley et al. [23]	2018	Urine	766		0.9	12	20.1	50.1	4.1	20.0 (CXM) 13.2 (CEP) 11.1 (CAZ) 15.9 (CRO)	22.7	21.8	NA	NA	NA	32.7	NA
Austria																	
AURES [24]	2016	Blood			5.7 °		50.5 *		NA	9.2 ∞	19.8				NA	NA	NA
AURES [24]	2020	Blood			6.4 °		46.1 *		NA	10.1 ∞	17.8				NA	NA	NA
USA																	
Kaye et al. [25]	2019	Urine	1 513 882		NA	NA	NA	NA	NA	3.2 (CEP) 11.9 (CEF)	21.1			NA	3.8	25.4	NA

AMK = Amikacin, AMP = Ampicillin, AMC = Amoxicillin + clavulanic acid, CXM = Cefuroxime, CFX = Cefotaxime, CAZ = Ceftazidim, CIP = Ciprofloxacin, CEP = Cefepime, CEF = Cefazoline, CRO = Ceftriaxone, GEN = Gentamicin, LEV = Levofloxacin, NAL = Nalidixic acid, NFN = Nitrofurantoin, NOR = Norfloxacin, PIP = Piperacillin, SXT = Trimethoprim + Sulfamethoxazole, TET = Tetracycline; NA = not available; * = Aminopenicillins; ∞ = third-generation Cephalosporins; ° = GEN and Tobramycin.

## Data Availability

Data can be provided by the authors.

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
