# Peer review of "The Antimicrobial Resistance (AMR) Rates of Uropathogens in a Rural Western African Area—A Retrospective Single-Center Study from Kpando, Ghana"

_antibiotics, 2022, doi:10.3390/antibiotics11121808_

Round 1
Reviewer 1 Report
The manuscript entitled “The antimicrobial resistance (AMR) rates of uropathogens in a rural Western African area - a retrospective single-center study from Kpando, Ghana” aimed to reveal the antimicrobial resistance (AMR) situation of uropathogens in Western Africa. It is a retrospective study of urine cultures from the rural Hospital in Ghana in two years period (2019-2021), but there is some restrictions in this study.
The lack of a clinical microbiologist who should evaluate the urine cultures and antimicrobial resistant patterns according to current guidelines is a huge restriction of the sutdy. For example; there is not any information about the isolated microorganisms from urine cultures whether they are really pathogens or only contaminants.
The antimicrobial resistant patterns should be evaluated based on guidelines from the Clinical and Laboratory Standards Institute at least 2019, not 2015. Some bacterial species are instrinsically resistant to some of the evaluated antibiotics in the study, so their antimicrobial susceptibility testing should not be performed.
In discussion section, the data with existing literature on E. coli AMR rates in Ghana were compared with the Austrian resistance report from 2020. To compare data sets from different origin of specimens and study population is meaningless for the literature.
The article has defects, research not conducted correctly and so it is unacceptable in present form.
Author Response
Thank you very much for your review. Please find attached our answers and the revised manuscript with track changes.
Kind regards
The authors

Reviewer 2 Report
I have reviewed the manuscript entitled "The antimicrobial resistance (AMR) rates of uropathogens in a rural Western African area - a retrospective single-center study from Kpando, Ghana" by Deininger et al.
The Authors just collected 348 MSU samples then investigated by microscopy and 1 microliter of the sample was cultured on CLED agar. After that the recovered colonies were identified and antimicrobial sensitivity testing was done by disk diffusion method.
Minor comments
1- Long period time (2 years and 3 months) was required to Collect urine samples
2- Some important clinical data of the patients are missing.
3- Table 1 is not essential.
4- The authors must discriminate between in- or out-patients and represent results as two groups.
5-No references were mentioned under Bacterial identification and count.
6- The methods of identification is not clear. Please write it briefly.
7-Please explain why sulf/trimethoprime and fosfomycin were not tested. In spite their importance in treatment of UTI "Other ABs (trimethoprim plus sulfamethoxazole [SXT] and fosfomycin) were not tested as standard and were therefore removed from the evaluation".
8-Although 348 urine samples were examined while 125 (35.9%) showed bacterial growth. Could you explain why 223 samples were the negative culture.
9- Table 3 is not well presented. Does the authors count intermediate as resistance or susceptible?
10- Figure 3 is not important. The results are represented in table 2.
Author Response

(The authors gave the same response as above.)

Reviewer 3 Report
1. abstract - it is not usual to start a sentence with a number. Also, add an aim to the background. Latin names should be in italic.
2. the paragraph "The distance to the Ghanaian capital Accra is 192 km, but due to bad road conditions, a trip to..." - should be in the methods section
3. Exclution?
4. where is the name of table 1? also, bullets are not needed
5. were male in 52.6% (n=180) and female in 47.4% (n=162). sentence is not finished
6. please add titles of all figures and tables
7. two brown colours in table 3 are hard to be differentiated
8. tables should be in the restults section and there is no need for table of abbreviations as you define them on first mentioning
9. add limitation section
Author Response

(The authors gave the same response as above.)

Round 2
Reviewer 1 Report
I appreciate the changes you made in the manuscript after my recommendations but in your study, an evaluation and a comment of a clinical microbiologist at least about antimicrobial susceptibility patterns of uropathogens is needed. It is also necessary to denote it in the “acknowledgments” section.
Author Response
Dear Reviewer,
thank you very much for your contributions and comments.
Best regards
The authors

Reviewer 2 Report
I have re-reviewed the manuscript.
All the comments were done but some of them were not clear. For example the identification of microorganisms, CLSI 2015 is not updated.
Author Response

(The authors gave the same response as above.)

Reviewer 3 Report
I believe this manuscript Is now ready for publication
Author Response

(The authors gave the same response as above.)
